# Bidirectional regulation of glial potassium buffering – glioprotection versus neuroprotection

Hailun Li[1], Lorenzo Lones[1], Aaron DiAntonio[1,2]*

[1]Department of Developmental Biology, Washington University in St. Louis School of Medicine, St. Louis, United States; [2]Needleman Center for Neurometabolism and Axonal Therapeutics, St. Louis, United States

**Abstract** Glia modulate neuronal excitability and seizure sensitivity by maintaining potassium and water homeostasis. A salt inducible kinase 3 (SIK3)-regulated gene expression program controls the glial capacity to buffer $K^+$ and water in *Drosophila*, however upstream regulatory mechanisms are unknown. Here, we identify an octopaminergic circuit linking neuronal activity to glial ion and water buffering. Under basal conditions, octopamine functions through the inhibitory octopaminergic G-protein-coupled receptor (GPCR) OctβR to upregulate glial buffering capacity, while under pathological $K^+$ stress, octopamine signals through the stimulatory octopaminergic GPCR OAMB1 to downregulate the glial buffering program. Failure to downregulate this program leads to intracellular glia swelling and stress signaling, suggesting that turning down this pathway is glioprotective. In the *eag shaker Drosophila* seizure model, the SIK3-mediated buffering pathway is inactivated. Reactivation of the glial buffering program dramatically suppresses neuronal hyperactivity, seizures, and shortened life span in this mutant. These findings highlight the therapeutic potential of a glial-centric therapeutic strategy for diseases of hyperexcitability.

*For correspondence:
diantonio@wustl.edu

**Competing interests:** The authors declare that no competing interests exist.

## Introduction

$K^+$ homeostasis in the nervous system is required to maintain a healthy level of neuronal activity. Neurons release $K^+$ ions as they repolarize during action potentials. If this $K^+$ builds up in the extracellular space, it can disrupt neuronal firing and set the stage for neuronal hyperexcitability. Conventional treatments for conditions of hyperexcitability mostly target neuronal channels. However, more than one-third of epilepsy patients suffer from medically intractable seizures, while others experience debilitating side effects as neuron-targeting treatments often interfere with healthy neuronal functions (*Chen et al., 2018*). Hence, there is an urgent need for innovative therapeutic approaches that improve seizure control. Here, we explore glial mechanisms that regulate $K^+$ balance to modulate neuronal excitability and seizure sensitivity.

Glia buffer $K^+$ stress by taking in excess $K^+$ ions and osmotically obliged water molecules from the extracellular space. This function allows glia to control the extracellular level of $K^+$ and thereby modulate neuronal excitability, raising the hope that glia may be targeted to restore $K^+$ homeostasis and suppress hyperexcitability in epilepsy and stroke (*Hübel and Ullah, 2016*; *Sontheimer, 1994*). Glial-centric therapeutic strategies for these conditions have not been possible, largely due to a lack of understanding of mechanisms controlling the glial $K^+$ buffering capacity. We previously discovered a signal transduction pathway that controls the glial capacity to buffer $K^+$ and water in *Drosophila* (*Li et al., 2019*). SIK3 (salt inducible kinase 3) is an AMPK family kinase that sequesters histone deacetylase 4 (HDAC4) in the cytoplasm, thereby promoting myocyte enhancer-factor 2 (Mef2)-dependent expression of $K^+$ and water transport molecules, including the *Drosophila* orthologs of aquaporin-4 and SPAK, a kinase necessary for activating the $Na^+/K^+$ transporter NKCC1. This

glial program suppresses extracellular nerve edema and prevents neuronal hyperexcitability and seizure in *Drosophila*, suggesting that it plays an important role in glial maintenance of $K^+$ and water homeostasis and is required for a healthy level of neuronal activity. This glial program is best characterized in wrapping glia, which ensheathe axons in the *Drosophila* peripheral nervous system (PNS) and are akin to vertebrate non-myelinating Schwann cells (*Bittern et al., 2020*), and so are well positioned to regulate the ionic composition around PNS axons (*Li et al., 2019*).

While we have elucidated downstream effectors by which SIK3 regulates $K^+$ and water buffering, the upstream signals that work through the SIK3 pathway to control glial buffering capacity are unknown. In other contexts, SIK3 integrates signals from different signal transduction pathways, coupling extracellular signals to changes in cellular responses (*Choi et al., 2015*; *Wang et al., 2011*; *Wein et al., 2018*). One of these pathways is G-protein-coupled receptor (GPCR)-cyclic AMP (cAMP)-protein kinase A (PKA) signaling, in which PKA directly inhibits SIK3 activity (*Wang et al., 2011*). A better understanding of upstream signaling mechanisms that regulate the SIK3 pathway and extracellular cues that modulate glial $K^+$ buffering capacity will inform approaches to leverage this glial function for therapeutic benefits in diseases of hyperexcitability.

Here, we identify an octopaminergic circuit linking neuronal activity to glial $K^+$ and water buffering. Octopamine regulates the glial buffering program via GPCR-dependent regulation of PKA, which inhibits SIK3. At baseline, octopamine acts through the inhibitory octopaminergic GPCR OctβR to activate the glial SIK3 pathway to promote $K^+$ and water buffering and healthy levels of excitability. Under pathological $K^+$ stress, in contrast, octopamine functions through the stimulatory octopaminergic GPCR OAMB1 to inhibit SIK3 and downregulate the glial capacity to buffer $K^+$ and water. While loss of the SIK3 pathway leads to extracellular edema, constitutive activation of the pathway results in intracellular glial swelling and activation of stress signaling, suggesting that turning down this pathway protects glia. In *eag shaker*, a classic *Drosophila* hyperexcitable mutant with defective $K^+$ channels, the SIK3-mediated glial buffering program is turned off, likely as a self-protective mechanism for glia. However, reactivation of this glial $K^+$ and water buffering program dramatically suppresses neuronal hyperexcitability, seizures, and shortened life span in *eag shaker*. Therefore, the maintenance of a robust glial $K^+$ and water buffering program in response to extreme neuronal excitability is beneficial to the organism despite the risks posed to glial health. Taken together, this study identifies a neuromodulatory circuit that links neuronal activity to glial $K^+$ buffering and highlights the promise of a glial-centric therapeutic strategy that restores $K^+$ and water homeostasis for control of hyperexcitability in diseases such as epilepsy and stroke.

## Results

### G-protein/PKA signaling in glia regulates $K^+$ and water homeostasis

Glia modulate neuronal excitability by removing $K^+$ and water from the extracellular space. Our previous work identified SIK3 as a key regulator of glial $K^+$ buffering that suppresses extracellular fluid accumulation, neuronal hyperexcitability, and seizure sensitivity (*Li et al., 2019*). Here, we sought to identify upstream signaling pathway(s) that regulate SIK3 activity. In the *Drosophila* fat body, cAMP-dependent PKA may phosphorylate and inhibit SIK3 (*Choi et al., 2015*; *Wang et al., 2011*). We hypothesized that this signaling axis also functions in glia, and that PKA is a negative regulator of SIK3. If so, PKA activation in glia would phenocopy the SIK3 mutant phenotype and cause peripheral nerve extracellular edema which is visualized as discrete swellings, a readout of disrupted ionic and water homeostasis (*Leiserson et al., 2000*; *Li et al., 2019*). PKA exists as a tetramer of regulatory and catalytic subunits, in which the catalytic subunits are liberated from the regulatory subunits after cAMP binds. To test if activated PKA inhibits SIK3, we took two complementary approaches. First, we employed the pan-glial driver Repo-GAL4 to overexpress the catalytic subunit PKA-C1. Second, we knocked down the regulatory subunit PKA-R1, which should activate the endogenous catalytic subunit. Both manipulations phenocopied the loss of SIK3 in glia and induced dramatic localized edema in larval peripheral nerves, suggesting that PKA hyperactivity in glia disrupts ionic and water homeostasis (*Figure 1A–B*). Since PKA is often regulated by GPCR-cAMP signaling, we tested if overexpression of constitutively active Gαs also phenocopies SIK3 loss-of-function (LOF) in glia. Indeed, this manipulation recapitulates the characteristic nerve swellings (*Figure 1C*).

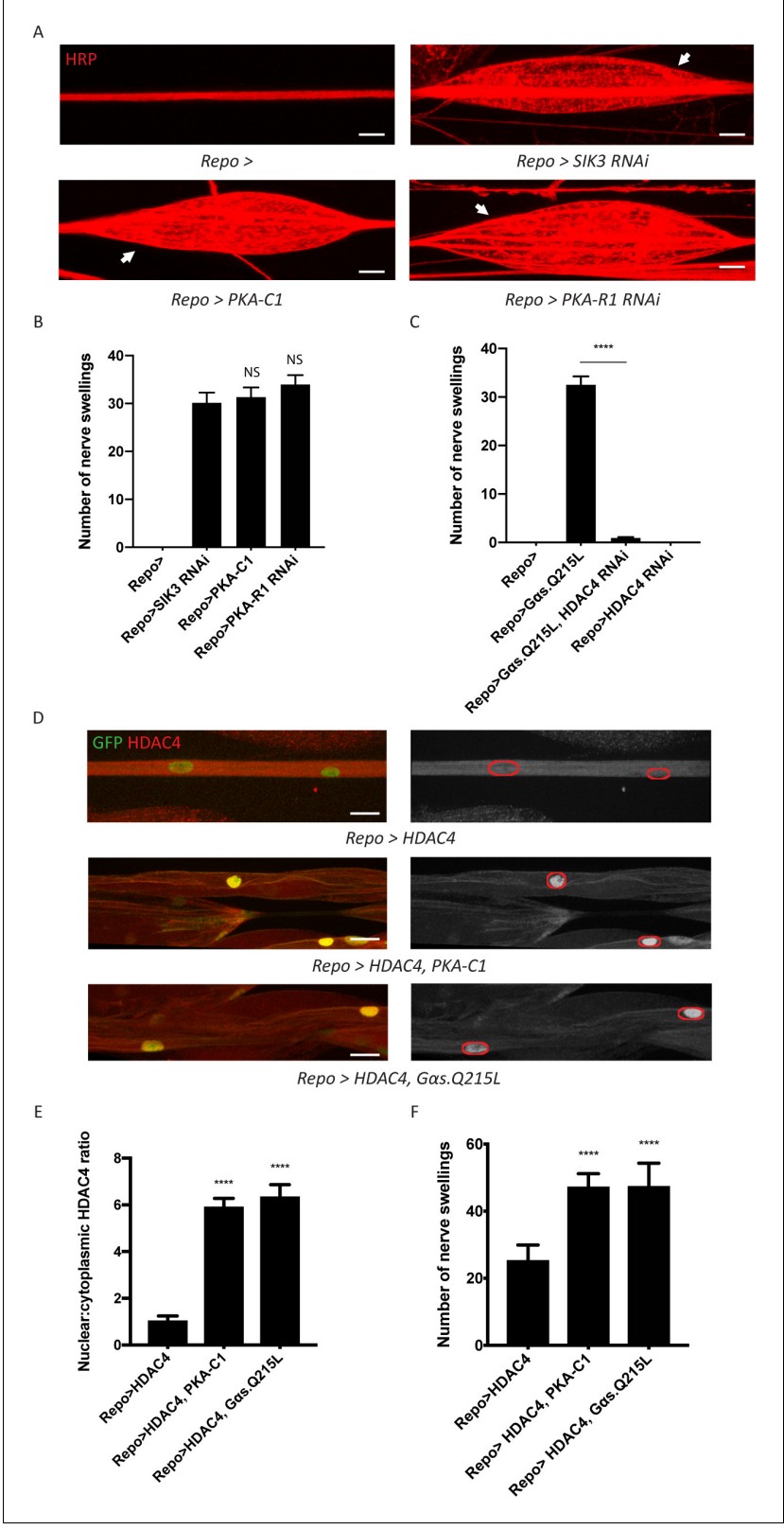

**Figure 1.** A G-protein-coupled receptor-protein kinase A (GPCR-PKA) signaling axis regulates salt inducible kinase 3 (SIK3)-mediated glial $K^+$ buffering. (**A**) Representative images of larval peripheral nerves stained for the nerve membrane marker HRP. Repo-GAL4, a pan-glial driver, was used to express LexA RNAi (control; abbreviated as *Repo>*), SIK3 RNAi (*Repo>SIK3 RNAi*), PKA-R1 RNAi (*Repo>PKA-R1 RNAi*), or a UAS-PKA-C1 transgene

*Figure 1 continued on next page*

*Figure 1 continued*

(*Repo>PKA-C1*). PKA hyperactivity, induced by either glial overexpression of its catalytic subunit (PKA-C1) or knockdown of a regulatory subunit (PKA-R1), causes localized nerve swellings (arrow) that resemble defects caused by loss of SIK3 from glia. Control larvae do not display swellings. Scale bars 20 µm. (B) Quantification of nerve swellings in (A). n ≥ 30. One-way ANOVA with Tukey's multiple comparisons; NS = not significant, p>0.05. (C) Quantification of nerve swellings per animal in control, larvae with glial expression of constitutively activated Gαs protein, HDAC4 RNAi, or co-expression of HDAC4 RNAi and Gαs. Constitutively activated Gαs in glia (*Repo>G_sα. Q215L*) causes nerve swellings that resemble (A); swellings are suppressed by loss of HDAC4 from glia (*Repo>G_sα. Q215L, HDAC4 RNAi*). Glial knockdown of HDAC4 (*Repo>HDAC4 RNAi*) does not result in nerve swellings. n ≥ 30. One-way ANOVA with Tukey's multiple comparisons; ****, p<0.0001. (D) Representative images of larval peripheral nerves demonstrating the effects of Gαs and PKA activation on HDAC4 localization in glia. Left: glial nuclei (green) and HDAC4 (red). Right: grayscale images show HDAC4 staining; glial nuclei are outlined. Scale bars 15 µm. (E) Quantification of nucleo:cytoplasmic ratio of HDAC4 for genotypes in (D). n ≥ 20. Data are presented as fold changes relative to *Repo>HDAC4*. One-way ANOVA with Tukey's multiple comparisons; ****, p<0.0001. (F) Quantification of number of nerve swellings per animal for genotypes in (D). n ≥ 20. One-way ANOVA with Tukey's multiple comparisons; ****, p<0.0001. Data are mean ± SEM.

We hypothesize that this G-protein/PKA signal regulates the SIK3 pathway to control glial $K^+$ and water homeostasis. If so, the induction of these nerve swellings should require HDAC4, the major downstream effector of SIK3 and inhibitor of the SIK3 signaling pathway. Indeed, glial-specific knockdown of HDAC4 fully suppresses the nerve swellings caused by Gαs overexpression (*Figure 1C*). In our previous work, we demonstrated that SIK3 sequesters HDAC4 in the glial cytoplasm, thereby promoting gene expression of a $K^+$ and water buffering program (*Li et al., 2019*). Here, we tested the model that Gαs and PKA also regulate HDAC4 localization to maintain glial $K^+$ homeostasis. We examined HDAC4 localization in glia by co-expressing a nuclear-localized GFP to label glial nuclei and a FLAG-tagged HDAC4 to measure HDAC4 intensity in both the nucleus and the cytoplasm. We then overexpressed PKA-C1 or Gαs in glia expressing FLAG-tagged HDAC4 (*Figure 1D–E*). In both cases, HDAC4 nuclear localization increases dramatically, resulting in an ~6-fold increase in the nucleo:cytoplasmic HDAC4 ratio. While glial overexpression of FLAG-tagged HDAC4 allows visualization of HDAC4 localization, it also causes HDAC4 hyperactivity in glia and induces nerve swellings. Consistent with the re-localization of HDAC4 into the nucleus induced by expression of Gαs and PKA-C1, these manipulations further increase the number of nerve swellings (*Figure 1F*). Since Gα-cAMP-PKA signaling is usually regulated by GPCRs, we conclude that a GPCR signaling pathway likely controls SIK3 activity to regulate $K^+$ and water homeostasis. Next, we sought to identify the ligand/receptor pair that regulates this glial $K^+$ and water buffering pathway.

## Octopamine signaling is required for $K^+$ and water buffering by glia

Studies of adipocytes have revealed a role for β-adrenergic signaling in cAMP-dependent regulation of SIK3 activity (*Berggreen et al., 2012*). Norepinephrine is a stress hormone that suppresses seizures in several mammalian models of epilepsy (*Mason and Corcoran, 1979*; *Szot et al., 1999*). The fly equivalent, octopamine, is a neuromodulator that regulates stress responses such as aggression and starvation resistance (*Suo et al., 2006*; *Zhou et al., 2008*). Here, we hypothesize that octopamine controls the glial capacity to buffer $K^+$ stress in the nervous system. First, we tested whether octopamine is required for SIK3-mediated $K^+$ buffering. Octopamine synthesis occurs in a well-defined small subset of neurons, in which tyrosine is processed into tyramine and subsequently into octopamine. This two-step process is catalyzed by tyrosine decarboxylase 2 (tdc2) and tyramine beta-hydroxylase (tβh), respectively. Vesicular monoamine transporters (VMATs) then package octopamine into secretory vesicles for later release. To test whether glial $K^+$ buffering is disrupted in mutants deprived of octopamine, we examined HDAC4 localization in glia expressing FLAG-tagged HDAC4, which provides a simple ON/OFF readout for the $K^+$ buffering program. HDAC4 nuclear localization increases dramatically in octopamine synthesis mutants, $tdc2^{R054}$ and $tβh^{nM18}$, suggesting that SIK3 is inhibited and so cannot sequester HDAC4 in the cytoplasm. These mutations also enhance the nerve swellings caused by HDAC4 hyperactivity in this glial background (*Figure 2C*). We observed similar results in dVMAT mutants (*Simon et al., 2009*), which cannot load octopamine into secretory vesicles and thereby have a defect in octopamine release (*Figure 2A–C*). Next, we tested the hypothesis that octopamine signals through GPCR-cAMP-PKA signaling for this function.

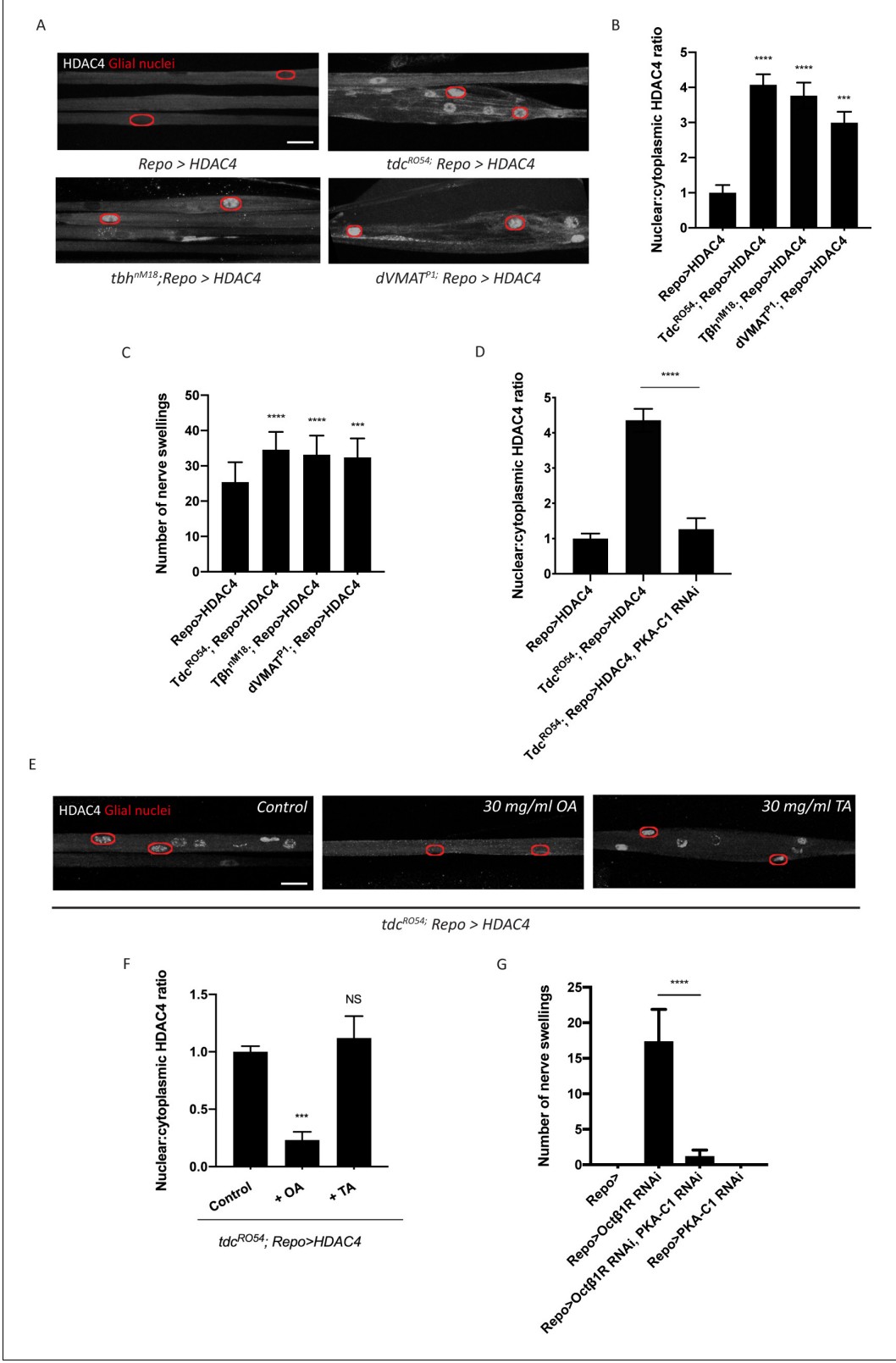

**Figure 2.** Octopamine is required to activate salt inducible kinase 3 (SIK3)-regulated K$^+$ buffering program in glia. (**A**) Representative images of peripheral nerves showing aberrant HDAC4 localization in octopamine synthesis and transport mutants. Grayscale images show HDAC4 staining; glial nuclei are outlined in red. Scale bars 15 μm. (**B**) Quantification of nucleo:cytoplasmic ratio of HDAC4 for genotypes in (**A**). n ≥ 20. Data are presented as fold

*Figure 2 continued on next page*

*Figure 2 continued*

changes relative to *Repo>HDAC4*. One-way ANOVA with Tukey's multiple comparisons; ***, p<0.001; ****, p<0.0001. (C) Quantification of number of nerve swellings per animal in HDAC4 overexpressing control larvae (*Repo>HDAC4*) and octopamine mutants. n ≥ 20 larvae per genotype. One-way ANOVA with Tukey's multiple comparisons; ***, p<0.001. ****, p<0.0001. (D) Quantification of HDAC4 localization in octopamine synthesis mutants, mutants with glial expression of LexA RNAi as control or PKA-C1 RNAi. Octopamine synthesis mutants (*tdc2^{R054}*) exhibit HDAC4 accumulation in glial nuclei; this nuclear localization is rescued by abolishing protein kinase A (PKA) catalytic activity in glia (*Repo>PKA-C1 RNAi*). One-way ANOVA with Tukey's multiple comparisons; ****, p<0.0001. (E) Representative images of nerves demonstrating the effect of 30 mg/ml octopamine or 30 mg/ml tyramine on octopamine synthesis mutant glia in an ex vivo assay. Scale bars 15 μm. (F) Quantification of HDAC4 nucleo:cytoplasmic ratio for genotypes in (D). Octopamine (30 mg/ml), but not TA (30 mg/ml) or KCl (500 mM), suppresses HDAC4 nuclear localization in glia. n ≥ 15. One-way ANOVA with Tukey's multiple comparisons; ***, p<0.001; NS = not significant, p>0.05. (G) Quantification of number of nerve swellings per animal in larvae with glial expression of LexA RNAi as control (*Repo>*), Octβ1R RNAi, PKA-C1 RNAi, or co-expression of Octβ1R RNAi and PKA-C1 RNAi. n ≥ 20. Student's t test; ****, p<0.0001. Data are mean ± SEM.

If so, octopamine regulation of HDAC4 localization should depend on PKA activity. Indeed, glial knockdown of PKA-C1 potently reverts HDAC4 nuclear localization in *tdc2^{R054}* mutants, pointing to PKA hyperactivity as the cause of HDAC4 mislocalization in octopamine synthesis mutants (*Figure 2D*).

Our findings demonstrate an essential role of octopamine in the activation of glial K$^+$ buffering. We hypothesize that this function is specific to octopamine and that octopamine acts directly on glia to regulate the SIK3 pathway. To test this model, we developed an ex vivo preparation to test if exogenous octopamine is sufficient to bypass the requirement for endogenous octopamine for the regulation of HDAC4 localization. In *tdc2^{R054}* mutants, which cannot synthesize either tyramine or octopamine, HDAC4 accumulates in the nucleus. We dissected third-instar larvae and treated the larval fillet for 2 min with physiological saline or saline containing 30 mg/ml octopamine hydrochloride or tyramine hydrochloride and assessed HDAC4 localization in glial nuclei. *tdc2^{R054}* mutants treated with control or tyramine-containing saline exhibit pronounced HDAC4 nuclear localization with a high nucleo:cytoplasmic HDAC4 ratio (*Figure 2E–F*). In contrast, 2 min exposure to octopamine induces rapid re-localization of HDAC4 to the cytosol. Hence, octopamine is sufficient to activate the SIK3 signaling pathway in glia, and the rapid time course strongly suggests that octopamine is acting directly on the glia.

If octopamine signals directly to glia, then there must be an octopamine receptor that mediates octopamine activation of glial K$^+$ buffering. Among the octopamine receptors that have been identified in *Drosophila*, OAMB, Octβ1R, Octβ2R, and Octβ3R can signal through cAMP as a second messenger (*El-Kholy et al., 2015*). Since octopamine is necessary for SIK3 signaling while PKA inhibits SIK3 signaling, we postulated that the relevant GPCR is coupled to an inhibitory Gα protein that normally suppresses PKA activity. Loss of this receptor from glia would result in PKA hyperactivity, inhibition of SIK3, and disrupted glial K$^+$ and water homeostasis. We performed RNAi-mediated knockdown of each candidate from above selectively in glia using Repo-GAL4 and assessed peripheral nerve morphology. From this screen, RNAi-mediated glial knockdown of Octβ1R results in the characteristic localized nerve edema phenotype that resembles SIK3 LOF mutants (*Figure 2G*). Moreover, glial knockdown of either PKA-C1 or HDAC4 potently suppresses nerve swellings in larvae deprived of Octβ1R in glia. These findings strongly support the model that octopamine functions through Octβ1R to inhibit PKA, thereby de-repressing SIK3 to sequester HDAC4 in the cytoplasm and promote proper K$^+$ and water buffering. Having demonstrated that octopamine normally activates the glial K$^+$ and water buffering program, we next investigated the impact of elevated octopamine on the glial SIK3 pathway.

## Excess octopamine inhibits the glial K$^+$ buffering program

In *Drosophila*, octopamine can activate distinct receptors that are localized in the same cell yet have antagonistic functions, potentially allowing for differential responses to low and high levels of octopamine (*Koon and Budnik, 2012*). We hypothesized that octopamine could have dual actions on glia to exert bidirectional control over K$^+$ buffering. We have demonstrated that octopamine functions

through the inhibitory GPCR Octβ1R to activate SIK3 signaling and upregulate glial K$^+$ and water buffering capacity (*Figure 2*). If octopamine indeed has dual effects on glial K$^+$ buffering, then octopamine acting through a stimulatory GPCR would enhance PKA activity in glia and thereby inhibit SIK3 signaling. As a first test of this model, we asked whether chronic treatment with elevated levels of octopamine can disrupt glial regulation of K$^+$ and water homeostasis. We increased octopamine levels by raising third instar larvae on a diet supplemented with octopamine hydrochloride or tyramine hydrochloride and examined peripheral nerve morphology. Wild-type larvae raised on a control diet do not develop swellings in their peripheral nerves (0 out of >100 larvae) (*Figure 3A*). A high tyramine diet also does not impact larval nerve morphology. However, a high octopamine diet increases the susceptibility of wild-type larvae to nerve edema, with ~70% of larvae developing swellings in peripheral nerves. Hence, high levels of octopamine disrupt the larval ability to maintain ionic and water homeostasis. These nerve swellings can be suppressed by enhancing the glial SIK3 signaling pathway, as PKA-C1 knockdown and HDAC4 knockdown selectively in glia both significantly reduce the frequency of nerve swellings in larvae raised on a high octopamine diet (*Figure 3A*). The suppression by PKA-C1 knockdown demonstrates that the effect of exogenous octopamine works through active PKA, supporting the model that high levels of octopamine work through a stimulatory GPCR.

To test directly whether high levels of octopamine modulate the SIK3 signaling pathway, we examined HDAC4 localization in glia expressing FLAG-tagged HDAC4 under control of the pan-glial driver Repo-Gal4. If exogenous octopamine downregulates the glial capacity to buffer K$^+$ by inhibiting SIK3, then this would lead to nuclear accumulation of HDAC4. Indeed, a high octopamine diet dramatically increases HDAC4 nuclear localization in glia (*Figure 3B–C*). This regulation of HDAC4 localization by octopamine depends on PKA activity, as glial knockdown of PKA-C1 completely abolishes octopamine-induced nuclear shuttling of HDAC4. We previously demonstrated that in the PNS this SIK3/HDAC4 pathway primarily regulates ion buffering in wrapping glia, the glial subtype that directly ensheathes the axon (*Li et al., 2019*). To assess whether octopamine regulation can occur in wrapping glia, we expressed FLAG-tagged HDAC4 under control of Nrv2-Gal4, which expresses exclusively in wrapping glia (*Sun et al., 1999*). With a high octopamine diet, HDAC4 in wrapping glia is present at much lower levels in the cytoplasm leading to a high nuclear:cytoplasmic ratio of HDAC4 (*Figure 3—figure supplement 1*). Hence, octopamine can inhibit the SIK3 pathway in the wrapping glia that primarily mediate ionic buffering around peripheral axons.

Having demonstrated that high levels of octopamine enhances PKA activity to inhibit SIK3 pathway in glia, we next searched for the relevant stimulatory GPCR in glia. OAMB is an octopamine receptor coupled to a Gαs protein that activates adenylyl cyclase to increase cAMP-PKA signaling (*Kim et al., 2013*). Glial knockdown of OAMB potently blocks HDAC4 nuclear localization in larvae raised on a high octopamine diet, demonstrating that high levels of octopamine act directly on glia through OAMB to inhibit the SIK3 pathway and turn down glial K$^+$ buffering. Hence, octopamine can bidirectionally regulate the glial capacity to buffer K$^+$ and water, and this differential effect works through distinct octopamine receptors and may depend on variations in the level of octopamine release.

Like its mammalian counterpart norepinephrine, octopamine is normally maintained at a low level and released as a stress signal during bouts of high neuronal activity or in response to high K$^+$ levels (*Orchard and Lange, 1987*). Here, we postulated that octopamine links neuronal activity to glial K$^+$ buffering, and that octopamine release correlates with the level of K$^+$ stress. If the effect of K$^+$ stress is mediated by octopamine signaling, then high K$^+$ stress should induce the same effects as high levels of octopamine and modulate HDAC4 localization in wild-type glia. To test this model, we raised larvae on a diet supplemented with 200 mM KCl and examined HDAC4 localization in their glia expressing FLAG-tagged HDAC4. Indeed, a high K$^+$ diet mimics a high octopamine diet and dramatically increases HDAC4 accumulation in the glial nuclei (*Figure 3D*). This regulation of HDAC4 localization is potently suppressed by glial knockdown of the octopamine receptor OAMB, demonstrating that sustained K$^+$ stress requires octopamine signaling to alter SIK3 pathway activity. Since K$^+$ stress is correlated with the level of neuronal excitability, these findings suggest that sustained changes in neuronal excitability function through differential octopamine release to bidirectionally modulate glial K$^+$ and water buffering capacity, likely by acting through receptors with antagonistic functions (*Figure 3E*).

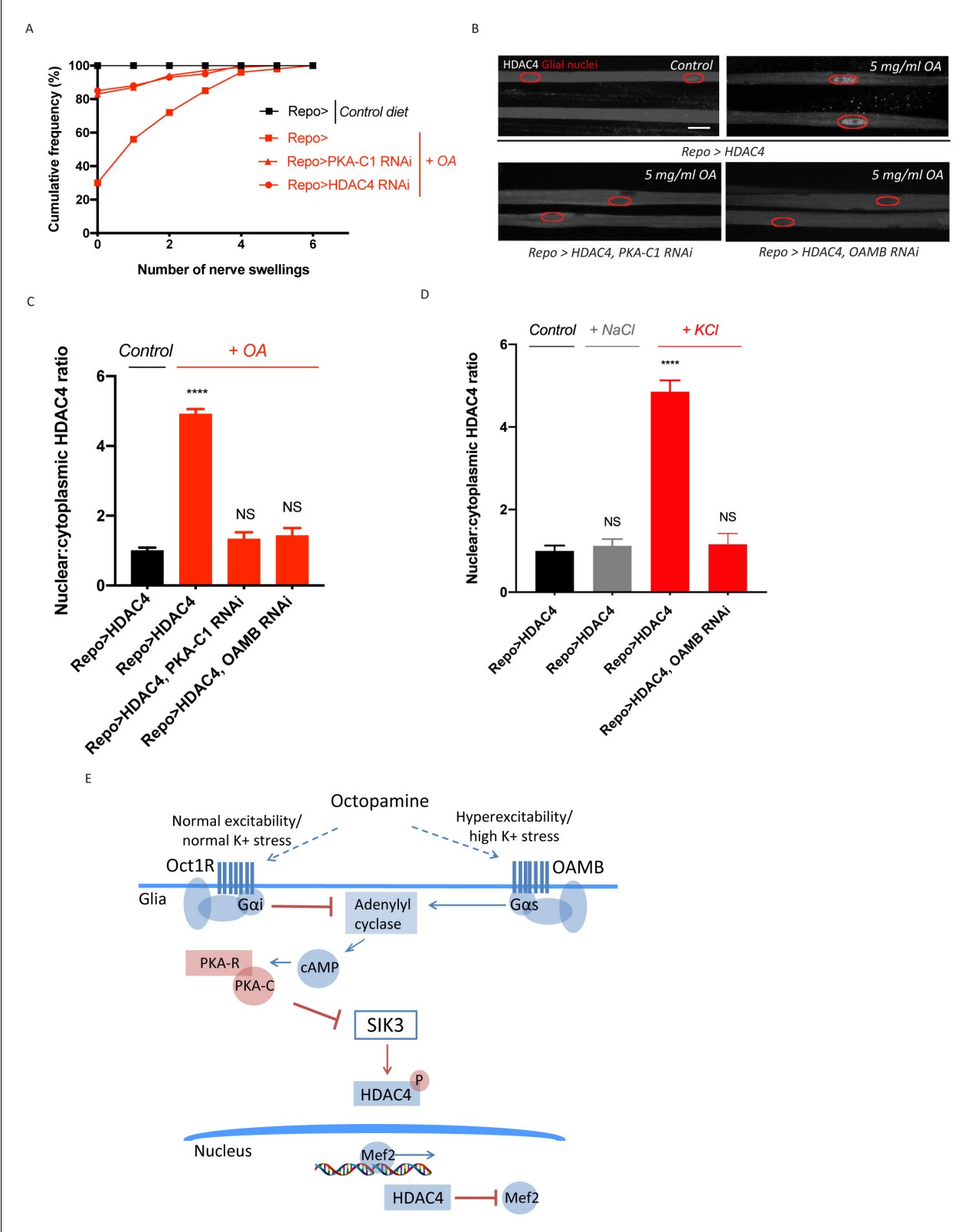

**Figure 3.** Excess octopamine inhibits salt inducible kinase 3 (SIK3) signaling to downregulate glial K$^+$ buffering. (**A**) Frequency of occurrence of nerve swellings in high octopamine-fed larvae with glial expression of LexA RNAi (*Repo>*), PKA-C1 RNAi, or HDAC4 RNAi. Larvae raised on control diet do not develop swellings; high octopamine (5 mg/ml) diet, but not high TA (5 mg/ml) diet, increases the percent of larvae exhibiting nerve swellings; this effect is blocked by loss of PKA-C1 or HDAC4 from glia. n ≥ 100. (**B**) Representative images of peripheral nerves demonstrating the effect of high
*Figure 3 continued on next page*

Figure 3 continued

octopamine diet on glial HDAC4 localization. High octopamine diet enhances HDAC4 nuclear localization in control larvae (*Repo>HDAC4*); this effect is blocked by glial knockdown of PKA-C1 (*Repo> PKA-C1 RNAi*) or OAMB (*Repo>OAMB RNAi*). Grayscale images show HDAC4 staining; glial nuclei are outlined in red. Scale bars 15 µm. High octopamine diet promotes nuclear localization of HDAC4 specifically in wrapping glia, as shown in **Figure 3—figure supplement 1**. (**C**) Quantification of HDAC4 nucleo:cytoplasmic ratio for genotypes in (**B**). n ≥ 30. Data are presented as fold changes relative to *Repo>HDAC4*. One-way ANOVA with Tukey's multiple comparisons; ****, $p<0.0001$. NS = not significant, $p>0.05$. (**D**) Quantification of HDAC4 localization in larvae raised on high salt diets. KCl-rich (200 mM) diet, but not NaCl-rich (200 mM) diet, promotes nuclear shuttling of HDAC4 in control larvae (*Repo>HDAC4*); this re-localization is inhibited by glial knockdown of OAMB (*Repo>OAMB RNAi*). n ≥ 20. One-way ANOVA with Tukey's multiple comparisons; ****, $p<0.0001$; NS, $p>0.05$. (**E**) Schematic model of octopamine exerting dual effects on SIK3-mediated glial $K^+$ buffering: in response to $K^+$ stress, different levels of octopamine act through receptors with antagonistic functions to differentially regulate SIK3-mediated glial $K^+$ buffering. Data are mean ± SEM.

The online version of this article includes the following figure supplement(s) for figure 3:

**Figure supplement 1.** Excess octopamine in wrapping glia downregulates glial $K^+$ buffering.

## Glia with enhanced $K^+$ and water buffering capacity undergo cellular swelling

Having demonstrated a seemingly counterintuitive relationship between high $K^+$ stress and reduced glial $K^+$ buffering capacity, we next explored the significance of this downregulation. We hypothesized that this mechanism protects glia from the negative effects of extreme $K^+$ stress (*Murphy et al., 2017*; *Risher et al., 2012*). To test this idea, we assessed consequences of a constitutively active glial $K^+$ buffering program. We independently knocked down two components that each would enhance SIK3 signaling in glia: (1) OAMB, the octopamine receptor that enhances PKA activity to inhibit SIK3 signaling, and (2) HDAC4, the central negative regulator of SIK3 signaling. First, we hypothesized that glia with enhanced $K^+$ buffering capacity would undergo cellular swelling as a result of taking in an excess load of $K^+$ ions and water molecules. *Drosophila* and mouse mutants with glial swelling exhibit an increase in nerve width along the entire length of their peripheral nerves (*Byun and Delpire, 2007*; *Rusan et al., 2014*). This is in contrast to the localized extracellular edema caused by SIK3 LOF in glia, which occurs in discrete locations along the nerve (*Figure 1A* and *Li et al., 2019*). Here, we raised larvae on control diet or a diet supplemented with 200 mM KCl and examined peripheral nerve morphology. Glial knockdown of OAMB results in a 'non-localized' nerve edema phenotype in both larvae raised on control diet and those on a high $K^+$ diet (*Figure 4A–B*). As expected, this edema phenotype is dramatically different than the extracellular nerve swellings in SIK3 mutants, as it occurs uniformly along the entire nerve. HDAC4 LOF in glia induces a similar edema phenotype, but only when larvae were provided with $K^+$ stress by being raised on a high $K^+$ diet. To directly test if this phenotype reflects edema inside the glia, we knocked down OAMB or HDAC4 in glia labeled with cytoplasmic RFP. Thin optical sections of nerves with OAMB or HDAC4 knockdown in glia revealed a large increase in intracellular glial cytoplasm that is indicative of increased glial volume (*Figure 4C*). This is in direct contrast to the appearance of the large extracellular swellings caused by glial loss of SIK3, which appear as large unstained regions surrounded by glial cytoplasm (*Figure 4C*). These results support our hypothesis that a constitutively active $K^+$ buffering program causes excess uptake of ions and osmotically obliged water and results in glial cellular swelling.

We next examined whether activation of the SIK3 pathway influences other aspects of glial health beyond cellular swelling. The c-Jun N-terminal kinase (JNK) is a classic stress kinase activated by a variety of cell stressors including the mechanical stress of cell stretching (*Pereira et al., 2011*). To test whether enhanced SIK3 signaling affects JNK activity in glia, we employed a transcriptional reporter of JNK pathway activity, a nuclear-localized lacZ enhancer trap inserted into *puckered (puc)*, a JNK phosphatase that is induced by JNK signaling as a negative regulatory mechanism. We expressed a nuclear-localized GFP in glia in the puc-lacZ background to label the glial nuclei. Basal expression of puc-lacZ in control larvae is relatively low, suggesting a minimal level of JNK activity in normal glia (*Figure 4D*). In contrast, SIK3 pathway activation by glial-specific knockdown of either OAMB or HDAC4 dramatically increases JNK activity, resulting in ~3-fold increase in puc-lacZ signal (*Figure 4D–E*). Taken together, we conclude that constitutive activation of the SIK3 $K^+$ and water buffering program causes intracellular glial swelling and activates a molecular stress response pathway. While these findings highlight the cell autonomous costs of an activated glial buffering

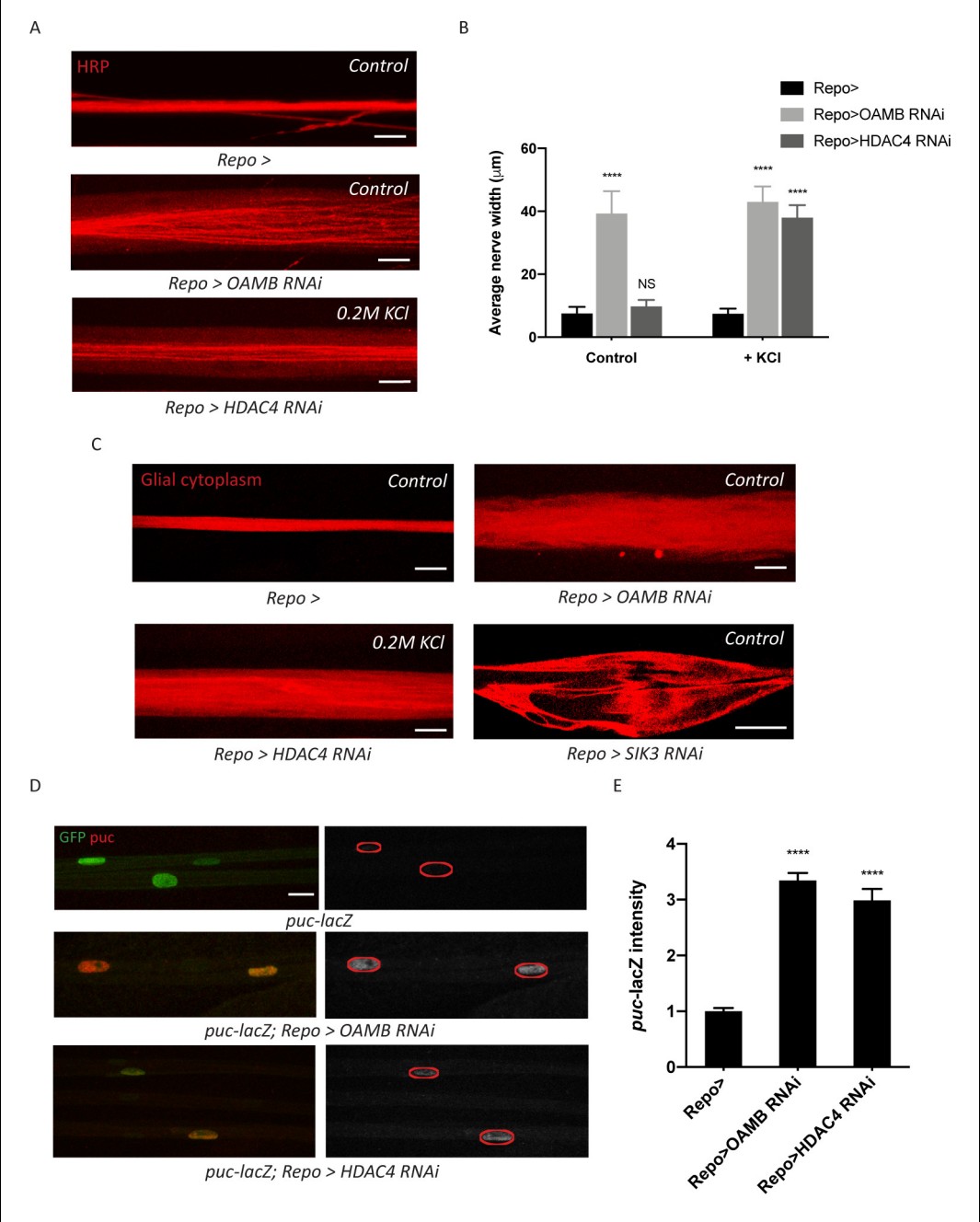

**Figure 4.** Glia with enhanced K$^+$ buffering capacity undergo swelling and stress responses. (**A**) Representative images of HRP-stained peripheral nerves demonstrating 'non-localized' edema in larvae with glial expression of LexA RNAi (*Repo>*), OAMB RNAi, or HDAC4 RNAi. Scale bars 20 μm. (**B**) Quantification of average nerve width for genotypes in (**A**). Loss of OAMB from glia causes 'non-localized' edema that results in a uniform increase in nerve width along the entire nerve; HDAC4 knockdown in glia induces a similar phenotype when larvae were fed a KCl-rich (200 mM) diet. n ≥ 15. Two-way ANOVA with Tukey's multiple comparisons; ****, p<0.0001; NS = not significant, p>0.05. (**C**) Representative thin optical sections of larval peripheral nerves with RFP-labeled (red) glial cytoplasm. (**D**) Representative images of larval peripheral nerves stained for c-Jun N-terminal kinase (JNK) pathway activity reporter *puc-lacZ*. Left: glial nuclei (green) and puc (red). Right: grayscale images show puc-lacZ staining; glial nuclei are outlined. Scale bars 15 μm. (**E**) Quantification of puc-lacZ signals in glia for genotypes in (**C**). Loss of OAMB or HDAC4 from glia dramatically increases *puc* expression in glia. n ≥ 20. One-way ANOVA with Tukey's multiple comparisons; ****, p<0.0001. Data are presented as fold changes relative to *puc-lacZ; Repo>*. Data are mean ± SEM.

program, we also tested whether enhanced K$^+$ and water buffering might provide organismal benefit in a seizure model.

## Activation of the glial buffering program suppresses hyperexcitability in a classic seizure mutant

We demonstrated that inducing K$^+$ stress by feeding larvae KCl re-localizes HDAC4 to the glial nuclei and thereby turns off the glial buffering program (*Figure 3E*). While high K$^+$ is a common proxy for hyperexcitability, we wished to test whether a bona fide hyperexcitable mutant would also turn down the glial K$^+$ and water buffering program. We employed a classic fly seizure model, *eag shaker (eag,Sh)*, which is mutant for two voltage-gated K$^+$ channels and displays dramatic neuronal hyperexcitability, seizures, and a very short life span (*Fergestad et al., 2006*; *Ganetzky and Wu, 1983*). To test if the *eag,Sh* mutant has altered glial SIK3 signaling, we examined HDAC4 localization in glia expressing FLAG-tagged HDAC4. In the *eag,Sh* mutant, there is greatly enhanced HDAC4 nuclear localization, demonstrating inhibition of the glial SIK3 pathway (*Figure 5A–B*). Moreover, loss of *eag* and *Shaker* exacerbates the localized nerve edema induced by HDAC4 hyperactivity in this glial background (*Figure 5C*). Hence, the glial buffering program is inhibited in this hyperexcitable mutant. Since we previously demonstrated that inhibition of the glial SIK3 pathway is sufficient to induce hyperexcitability and seizures (*Li et al., 2019*), we explored whether inhibition of the glial buffering program is contributing to the hyperexcitable phenotype in the *eag,Sh* mutant.

If glial defects in the K$^+$ and water buffering program contribute to *eag,Sh* phenotypes, then enhancing the glial buffering capacity in the *eag,Sh* mutant should ameliorate defects in K$^+$ buffering and suppress seizure phenotypes. To test this idea, we performed glial-specific RNAi-mediated knockdown of HDAC4 to enhance SIK3-mediated glial K$^+$ and water buffering in the *eag,Sh* mutant. We observed dramatic suppression of a series of *eag,Sh* phenotypes. First, *eag Sh* mutants exhibit down-turned wings, a phenotype commonly observed in *Drosophila* hyperexcitability mutants (*Figure 5D*). This morphological abnormality is suppressed by glial knockdown of HDAC4. In fact, inhibition of glial HDAC4 in these mutants results in an upheld wing phenotype, which occurs in mutants with muscle hypercontraction as a result of altered neuronal excitability (*Katti et al., 2017*; *Montana and Littleton, 2004*). Next, we tested if HDAC4 inhibition suppresses neuronal hyperexcitability in the *eag,Sh* mutant at the larval neuromuscular junction (NMJ). In this system, the peripheral nerve is severed such that motor axons do not receive input from their cell bodies and so do not fire action potentials without exogenous nerve stimulation. When recording from postsynaptic muscles of wild-type larvae, only miniature excitatory junction potentials (mEJPs) generated by the spontaneous release of single vesicles are detected (*Figure 5E*). In contrast, *eag,Sh* displays not only mEJPs, but also much larger evoked junction potentials (EJPs) that result from the spontaneous firing of axons. These observations are consistent with findings from previous studies on *eag,Sh* (*Ganetzky and Wu, 1983*) and are a hallmark of neuronal hyperexcitability. Activation of the SIK3 glial buffering program in *eag,Sh* mutants by selective glial knockdown of HDAC4 leads to an ~60% reduction in the frequency of these spontaneous EJPs (*Figure 5E–F*). Since enhancing glial K$^+$ buffering potently suppresses the neuronal hyperexcitability induced by loss of the *eag* and *Shaker* potassium channels, the hyperexcitability attributed to the absence of these channels is due in part to the lack of sufficient glial ion and water buffering.

We next tested whether enhancing glial K$^+$ and water buffering could suppress seizures in the *eag,Sh* mutant (*Kuebler et al., 2001*; *Pavlidis and Tanouye, 1995*). We assessed bang-sensitive seizures by delivering a 15 s mechanical shock and recording the frequency of seizing behaviors including shuddering, leg shaking, contorted posturing, and spinning flight. Consistent with previous findings, *eag,Sh* mutants are about 25 times more likely to seize than control flies (*Figure 5H*). They also have much longer seizures, which last for up to a minute while in control flies seizing behavior lasts only a few seconds. Glial-specific inhibition of HDAC4 dramatically suppresses the seizure behavior of *eag,Sh* mutants, reducing the incidence of seizure by ~80% and seizure duration to a few seconds. Finally, we tested whether enhancing glial K$^+$ buffering capacity improves the life span of *eag,Sh* mutants. *eag,Sh* mutants have a dramatically shortened life span compared to control flies (*Figure 5I* and *Fergestad et al., 2006*). Remarkably, glial-specific knockdown of HDAC4 in the *eag, Sh* mutant results in a dramatic extension of life span to essentially control levels. Taken together, these findings demonstrate that enhancing glial mechanisms that boost K$^+$ and water homeostasis

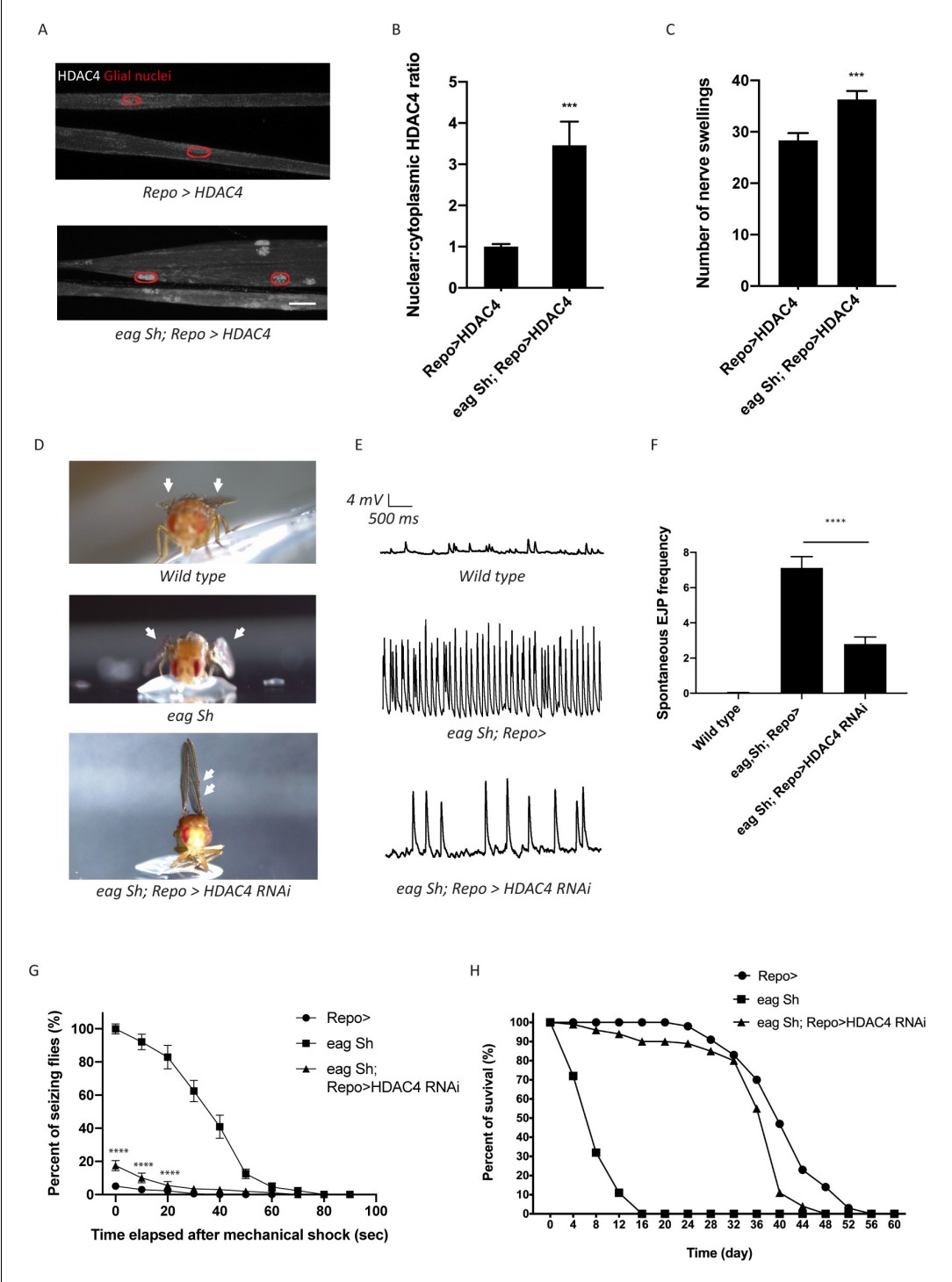

**Figure 5.** Enhanced glial K$^+$ buffering suppresses hyperexcitability in *eag,Sh* mutant. (**A**) Representative images of peripheral nerves demonstrating aberrant HDAC4 localization in *eag,Sh* mutant. Grayscale images show HDAC4 staining; glial nuclei are outlined in red. Scale bars 15 μm. (**B**) Quantification of HDAC4 nucleo:cytoplasmic ratio for genotypes in (**A**). n ≥ 20. Data are presented as fold changes relative to *Repo>HDAC4.* Student's t test; ***, p<0.001. (**C**) Quantification of nerve swellings in HDAC4-overexpressing control and eag shaker mutant. n ≥ 20. Student's t test; ***, p<0.001. (**D**) Representative images of wing morphology in control, *eag,Sh* mutants, and mutants with glial knockdown of HDAC4. *Eag,Sh* mutants exhibit down-turned wings (arrow) that are not observed in control flies; this phenotype is suppressed by glial-specific inhibition of HDAC4 (*Repo>HDAC4 RNAi*). (**E**) Representative physiological traces recorded from larval neuromuscular junctions (NMJs) for genotypes in (**D**). Control larvae (*Repo>*) only exhibit miniature excitatory junction potentials (mEJPs); *eag,Sh* exhibit spontaneous evoked junction potentials (EJPs). These spontaneous EJPs are suppressed by glial expression of HDAC4 RNAi. (**F**) Quantification of frequency of spontaneous EJPs for genotypes in (**D**). A minimum of 50

*Figure 5 continued on next page*

Figure 5 continued

consecutive events were analyzed over a passive recording window (up to 75 consecutive events or 120 s, whichever happened first), and events with amplitudes ≥4 mV were considered as spontaneous EJPs. n = 8 for *Repo>*; n = 8 for *eag,Sh*; n = 8 for *eag,Sh; Repo>HDAC4 RNAi*. One-way ANOVA with Tukey's multiple comparisons; p<0.0001. (G) Time course of vortex-induced seizure behaviors for genotypes in (D). n ≥ 5 groups of 10 flies per genotype per time point. Two-tailed Student's t test; ****, p<0.0001. (H) Life span analysis of genotypes in (D). n ≥ 50. Data are mean ± SEM.

profoundly suppresses electrophysiological, behavioral, and survival phenotypes of *eag,Sh*, the paradigmatic *Drosophila* neuronal hyperexcitability mutant.

## Discussion

$K^+$ dysregulation drives edema and neuronal hyperexcitability and can lead to brain damage in patients with epilepsy and stroke. Glia buffer $K^+$ stress by taking up excess $K^+$ ions and water to maintain a healthy extracellular environment for neuronal excitability. We previously identified SIK3 as the central node of a signal transduction pathway that controls the glial capacity to buffer $K^+$ and water in *Drosophila* (*Li et al., 2019*). Here, we uncovered upstream regulatory mechanisms controlling the SIK3 pathway and demonstrated that activation of this glial program can dramatically ameliorate the pathological consequences of neuronal hyperexcibiltiy. An octopaminergic circuit couples neuronal activity to glia buffering, exerting a dual effect on SIK3-mediated $K^+$ buffering: low levels of octopamine enhance the glial $K^+$ buffering capacity, while high levels of octopamine inhibit this buffering program, likely to protect glia against extreme $K^+$ stress. Glial-specific inhibition of HDAC4, a central repressor of SIK3 signaling, results in a constitutively active glial buffering program that dramatically suppresses seizure and extends life span in a classic epilepsy model. Hence, augmenting glial $K^+$ and water buffering holds promise as a therapeutic approach for the treatment of seizures.

### Glial SIK3 signaling controls cell volume regulation

The glial SIK3/HDAC4/Mef2 pathway regulates expression of both fray/SPAK, a kinase controlling ion transporter activity, and aquaporin 4, a water channel essential for volume regulation (*Leiserson et al., 2011*; *Li et al., 2019*; *Papadopoulos et al., 2004*). Our original analysis of SIK3 LOF mutants demonstrated that glial SIK3 inhibits extracellular edema, presumably by promoting the uptake of ions and water into glial cells. This is an essential activity of glia, since accumulation of $K^+$ in the extracellular space around axons leads to hyperexcitability and heightened seizure susceptibility. Since peripheral nerves must continuously fire, this raised the question of why expression of fray and aquaporin 4 would be regulated by SIK3, which is best understood to integrate opposing signals such as feeding and fasting to control the levels of downstream transcriptional targets (*Choi et al., 2015*). If ion and water buffering were so essential, then why is expression of essential buffering proteins not constitutive? Our findings suggest that the downregulating glial $K^+$ buffering capacity is a protective response that prevents pathological glial swelling. Activity-dependent glial swelling is a commonly observed physiological phenomenon, but when taking in an excess load of ions and water, glia may swell to the point of cytolysis (*MacVicar et al., 2002*). We have shown that glia with enhanced $K^+$ buffering capacity undergo cell swelling that is exacerbated by extreme $K^+$ stress. These glia also exhibit enhanced JNK pathway activity, which responds to a variety of stressors, including mechanical stress caused by cell stretching (*Pereira et al., 2011*). JNK plays a critical role in the regulation of apoptosis, suggesting that pathological swelling that results from enhanced $K^+$ buffering could promote cell death (*Dhanasekaran and Reddy, 2008*). Taken together, these findings demonstrate that overactivity of glial SIK3 activity leads to intracellular edema while underactivity of glial SIK3 signaling leads to extracellular edema. Hence, SIK3 balances the need for robust extracellular buffering with the health of the glial cell. We suggest that the ability of the SIK3 pathway to integrate disparate upstream signals is important for maintaining this balance between an extracellular milieu conducive to neuronal firing with intracellular volume regulation that maintains glial health.

## Dual actions of octopamine in controlling glial K$^+$ buffering

Norepinephrine is a stress hormone that can exert anticonvulsant effects in the mammalian nervous system (*Szot et al., 1999*). Moreover, norepinephrine regulates astrocyte transporter activity (*Monai et al., 2019*). In *Drosophila*, octopamine is structurally and functionally analogous to norepinephrine and regulates a variety of stress responses, including aggression, alertness, and starvation (*Crocker and Sehgal, 2008*; *Suo et al., 2006*; *Zhou et al., 2008*). Octopamine is also similar to norepinephrine in that it exerts dual effects on cellular responses by signaling through receptors with antagonistic functions (*Koon and Budnik, 2012*; *Papaefthimiou and Theophilidis, 2011*). In this study, we propose that octopamine bidirectionally controls SIK3-mediated glial K$^+$ buffering in response to changes in neuronal excitability.

Like its mammalian counterpart, octopamine is normally maintained at low levels and is released upon physiological stimuli to help the system mount a stress response. Octopamine release can be enhanced by a surge in the level of extracellular K$^+$, suggesting that its concentration correlates with the level of K$^+$ stress in the nervous system (*Orchard and Lange, 1987*). Here, we propose a model in which octopamine release links activity-dependent K$^+$ stress to glial K$^+$ buffering. This model is supported by a number of key findings. First, octopamine is required for SIK3 signaling and this regulation occurs through the inhibitory octopaminergic GPCR Octβ1R. Second, elevated levels of either octopamine or K$^+$ each inhibit the glial SIK3 pathway, and both of these effects require the stimulatory octopaminergic GPCR OAMB. Hence, elevated K$^+$ works through octopaminergic signaling to regulate the SIK3 pathway and glial buffering. Finally, the glial SIK3 pathway is inhibited in the hyperexcitable *eag,Sh* mutant. We propose that basal levels of octopamine promote SIK3-dependent glial K$^+$ buffering to cope with physiological K$^+$ stress and maintain a healthy level of neuronal activity. When neurons become hyperexcitable, as would occur under pathological conditions, levels of extracellular K$^+$ rise and more octopamine is released. High levels of octopamine will inhibit SIK3 activity and turn down glial K$^+$ buffering, likely to protect glia from pathological cell swelling.

This model posits that different levels of octopamine differentially activate Octβ1R and OAMB to exert these dual effects on glial K$^+$ buffering. Octβ1R couples to the inhibitory Gαi protein, whereas OAMB can function through the excitatory Gαs protein (*Kim et al., 2013*; *Koon and Budnik, 2012*). How might octopamine differentially signal through both Octβ1R and OAMB when they bind to the same ligand, reside in the same cell, and have antagonistic functions? Potentially, Octβ1R and OAMB could have different binding affinities for octopamine, with Octβ1R having a higher affinity such that Octβ1R is preferentially activated at low levels of octopamine. Inhibitory Gαi-coupled Octβ1R would decrease cAMP levels and inhibit PKA activity, thereby relieving the inhibition on SIK3 to upregulate glial K$^+$ buffering capacity. Conversely, octopamine would bind to the lower affinity OAMB when octopamine levels are high, as would occur during hyperexcitability. High levels of octopamine signaling through the excitatory Gαs-coupled OAMB would enhance PKA activity and downregulate the glial capacity to buffer K$^+$ stress. Hence, different binding affinities would allow Octβ1R and OAMB to be activated by different octopamine concentrations. A similar differential regulation via opposing octopaminergic receptors residing in the same cell is described at the larval NMJ, where octopamine signals through inhibitory Octβ1R and excitatory Octβ2R to bidirectionally control synaptic function (*Koon and Budnik, 2012*).

We have defined an octopaminergic circuit that links neuronal activity to glial K$^+$ buffering to maintain K$^+$ homeostasis and neuronal excitability. Neuromodulators such as octopamine are useful for integrating synaptic activity over time and space rather than signaling rapidly and locally like classical neurotransmitters. Moreover, altering the glial transcriptional response will be reasonably slow to change physiological function since such changes require transcription and translation. Hence, this system is likely tuned to respond to average neuronal activity rather than to transient changes in neuronal firing. This pathway may work in concert with additional mechanisms for rapid tuning of glial ion transporter function as described in mammalian astrocytes (*Bellot-Saez et al., 2017*).

This model for octopaminergic regulation of the glial SIK3-dependent ionic buffering program leaves open a number of important questions. First, is the octopamine released locally or does release from the central octopaminergic neurons diffuse widely to set the glial tone for ionic buffering? Second, while we showed that a glial octopamine receptor is necessary for pathway modulation by octopamine, it would also be interesting to test whether direct activation of octopaminergic neurons is sufficient to regulate the glial SIK3 pathway. Similarly, is direct inhibition of octopaminergic

neurons capable of rescuing the whole animal phenotypes of the *eag, Sh* mutants? Third, while we have focused on octopaminergic regulation, in other systems the SIK3 pathway is regulated by additional upstream signals including the kinase Lkb1 and the insulin receptor (*Choi et al., 2015*). Input from these additional pathways would allow for regulation of glial ion and water buffering in locations that may not be accessible for octopaminergic neuromodulation. Finally, we have focused our analysis on peripheral nerves and peripheral glia. In future studies it will be important to assess the role of this SIK3 buffering program in the cortex glia that buffer ions around neuronal cell bodies.

## Enhancing glial K$^+$ buffering can dramatically suppress neuronal hyperexcitability

With hyperexcitability, waves of action potentials can lead to increases in extracellular K$^+$ that, if not properly buffered, will lead to neuronal depolarization and further hyperexcitability (*Kofuji and Newman, 2004*). We explored the role of glial buffering in a paradigmatic *Drosophila* seizure mutant, *eag,Sh*, that exhibits dramatic neuronal hyperexcitability, seizure behaviors, and shortened life span (*Fergestad et al., 2006*; *Ganetzky and Wu, 1983*). Although we would expect *eag,Sh* larvae to have an increased need for glial buffering, we found that the glial SIK3 signaling pathway is downregulated, likely as part of the glial protective response described above. Since downregulating the glial SIK3 pathway is sufficient to cause hyperexcitability and seizures, we tested whether this inhibition of SIK3 could be exacerbating the *eag,Sh* phenotypes. HDAC4 is the key repressor of SIK3-regulated glial K$^+$ buffering, and so we activated SIK3 signaling by glial knockdown of HDAC4. Reactivation of the glial buffering program dramatically suppressed neuronal hyperexcitability, seizures, and shortened life span in *eag,Sh*. This result is quite remarkable, since the neurons are still deficient for two extremely important K$^+$ channels. Nonetheless, the pathological consequences for the organism are almost wholly mitigated by enhanced glial buffering. This highlights the potential of glial-centric therapeutic strategies for diseases of hyperexcitability.

## Materials and methods

### *Drosophila* stocks and genetics

Fly stocks were maintained at room temperature using standard techniques. Experimental crosses were raised at 25°C in 60% relative humidity. The following flies were used in this study: UAS-HDAC4 (a gift from Biao Wang); *tdc2$^{R054}$* and *tbh$^{nM18}$* are gifts from Vivian Budnick; *dVMAT$^{P1}$* (a gift from David Krantz); puc-lacZ; UAS-mito-GFP; *eag$^1$* shaker$^{120}$ (a gift from Troy Littleton). Flies obtained from the Bloomington *Drosophila* Stock Center include Repo-GAL4, Nrv2-Gal4, UAS-LexA RNAi (RRID:BDSC_67945), UAS-PKA-C1 (RRID:BDSC_35554), UAS-PKA-R1 RNAi (RRID:BDSC_27308), UAS-G$_s$α. Q215L (RRID:BDSC_6490), UAS-nls-GFP (RRID:BDSC_4775), UAS-HDAC4 RNAi (RRID:BDSC_28549, BDSC_34774), UAS-Octβ1R RNAi (RRID:BDSC_31107), and UAS-OAMB RNAi (RRID:BDSC_31171). UAS-SIK3 RNAi (KK 106268) flies were obtained from the Vienna *Drosophila* Resource Center. In GAL4 experiments, unless otherwise noted, control flies were generated by crossing virgin females with the GAL4 driver to UAS-LexA RNAi male flies.

### Immunocytochemistry

Third instar larvae were dissected in phosphate-buffered saline (PBS) and immediately fixed with either Bouin's fixative for 10 min or 4% paraformaldehyde for 20 min at room temperature. Larval preps were then washed 3× for 10 min with PBS + 0.1% Triton X-100 (PBST) and blocked for 30 min with PBST + 5% goat serum. To assess peripheral nerve morphology, larvae were incubated in Cy3-conjugated goat α-HRP antibody (1:1000; Jackson ImmunoResearch, cat# 123-165-021, RRID:AB 2338959) for 60 min at room temperature. In all other experiments, larvae were incubated overnight at 4°C in the following primary antibodies: mouse α-FLAG antibody (1:1000; Sigma, cat# F3165, RRID:AB_259529); rabbit α-FLAG (1:1000; Cell Signaling, cat# 147935); rabbit α-RFP (1:500; Clontech, cat# 632496, RRID:AB_10013483); mouse α-lacZ antibody (1:100; Developmental Studies Hybridoma Bank, cat# 40-1a, RRID:AB_2314509). On the following day, larvae were washed with PBST for 10 min 3× and incubated for 90 min at room temperature in the following secondary antibodies: Cy3 goat α-mouse (1:1000; Jackson ImmunoResearch, cat# 115-165-146, RRID:AB_2338680); Cy3 goat α-rabbit (1:1000; Invitrogen, cat# A-10520, RRID:AB_2534029); AlexaFluor 488-

conjugated goat α-rabbit (1:1000; Invitrogen, cat# A-11034, RRID:AB_2576217); 647 AffiniPure goat α-HRP (1:1000; Jackson ImmunoResearch, cat# 123-065-021). After 3 × 10 min wash in PBST, larvae were transferred to be equilibrated in 70% glycerol in PBS for at least 1 hr. Larvae were then mounted in Vectashield (Vector Laboratories) and ready to be imaged.

## Imaging and analysis
### Peripheral nerve morphology
Larvae peripheral nerves were stained for α-HRP and imaged using 20×/0.60 NA and 40×/1.15 NA oil immersion objectives on a Leica TCS SPE confocal microscope, a Leica DFC7000 T camera, and the LAS X software. We acquired images for all genotypes and conditions in the same experiment in one sitting using identical laser power, gain, and offset settings. Images shown are maximal Z-projections of confocal stacks, except for thin optical sections used to highlight glial swellings. Photoshop CC (2.2, Adobe) and Illustrator (15.0.0, Adobe) were used to minimally process the images in preparation for final figures. Nerve swellings were quantified by identifying and scoring regions with a maximum nerve width >15 μm. The total number of nerve swellings was measured for each larva and subsequently averaged for each genotype and condition.

### HDAC4 localization
Larvae were stained for FLAG-tagged HDAC4 in glia and GFP-labeled glial nuclei. For wrapping glia, glial nuclei were stained with mouse α-Repo antibody, HDAC4 was labeled with rabbit α-FLAG, and nerves were labeled with 647 α-HRP antibody. Their peripheral nerves were imaged using 40×/1.15 NA oil immersion objective on the confocal microscope and other equipment mentioned above. Images were acquired under identical settings and shown as maximal Z-projections of confocal stacks. Fluorescence intensity of FLAG-tagged HDAC4 was analyzed using Image J (1.52 k, National Institute of Health). The method used to quantify HDAC4 localization in glial nuclei and cytosol was described in a previous study. In brief, we used the GFP channel to generate a nuclear mask and applied this mask to the FLAG channel to selectively measure HDAC4 levels in glial nuclei. To assess HDAC4 levels in glial cytoplasm, FLAG signals in the 'non-nuclei' areas were also analyzed by subtracting the mask from total area of the peripheral nerve. For wrapping glia, we used the Repo channel to generate a nuclear mask and applied this to the FLAG signals in the 'non-nuclei' areas marked by HRP. The nuclear:cytoplasmic ratio of HDAC4 intensity was then calculated using these two measurements. A minimum of 60 independent peripheral nerves was examined for each genotype and condition and normalized to the control.

### JNK activity
puc-lacZ was used as a nuclear-localized transcriptional reporter of JNK pathway activity. Larvae expressing puc-lacZ were stained for lacZ and GFP-labeled glial nuclei, and their peripheral nerves were imaged using the methods described above. To quantify the expression level of puc-lacZ, glial nuclei were selected using the GFP channel and assessed on the intensity of lacZ signals. A minimum of 20 glial nuclei from different nerves were assessed for each genotype and then normalized to the control.

## High octopamine/high K$^+$ experiments
### Ex vivo assay
Before each experiment, octopamine hydrochloride (Sigma-Aldrich, cat# O0250) or tyramine hydrochloride (Sigma-Aldrich, cat# T2879) was resuspended from powder using HL3.1 solution with 0.35 mM Ca$^{2+}$ to reach a concentration of 30 mg/ml. Third instar larvae were dissected in HL3.1 solution and then immediately incubated in high octopamine or TA solution for 2 min at room temperature. Living larval preps were then rinsed with HL3.1 solution 3× before they were fixed in 4% paraformaldehyde. Larvae were further processed and assessed on HDAC4 localization using standard techniques as described above.

### Feeding assay
Octopamine hydrochloride or tyramine hydrochloride was resuspended using PBS and added to freshly made Formula 4–24 Instant *Drosophila* Medium Blue Food (Carolina, cat# 173210) to reach a

final concentration of 5 mg/ml. Experimental crosses were made on Blue Food such that larvae were always exposed to the drug until being assessed on peripheral nerve morphology or HDAC4 localization. In high salt experiments, larvae were raised on Blue Food supplemented with 200 mM KCl or NaCl. Control animals were larvae of the same genotype that were raised on Blue Food containing only PBS.

## Electrophysiology

Third instar larvae were dissected in $Ca^{2+}$-free HL3.1 buffer solution (70 mm NaCl, 5 mm KCl, 8 mm MgCl2, 10 mm NaHCO3, 5 mm trehalose, 5 mm HEPES, and 0 mm $Ca^{++}$, pH 7.2). Larval fillets were washed with and recorded in HL3.1 solution with 0.35 mM $Ca^{2+}$. Spontaneous EJPs and mEJPs were recorded from larval muscle 6 in segments A2, A3, and A4 using borosilicate sharp electrodes. In any given experiment, a minimum of 50 consecutive events were measured over a passive recording window. MiniAnalysis Software (Synaptosoft, Decatur, GA) was used to analyze $\leq 75$ consecutive events or a total recording window of 120 s, whichever happened first. All events were filtered by amplitude and events larger than 4 mV were classified as 'spontaneous EJPs'. The number of these events was divided by the exact length of recording to calculate the frequency of spontaneous EJPs. In all electrophysiology experiments, cells were chosen that had a resting membrane potential between $-60$ and $-80$ mV and muscle input resistance larger than 5 MΩ.

## Behavioral analysis

Adult virgin flies were collected within 24 hr post-eclosion and rested for 2 hr before being tested for bang sensitivity. They were transferred to an empty culture vial in groups of 5-10 and mechanically stimulated for ~15 s using a vortex mixer (Fisher Scientific). In the same vial, these flies were then assessed for seizure behaviors that lasted at least 1 s, including shuddering, leg shaking, spinning flight, and contorted posture. The number of flies seizing was recorded every 10 s for at least 2 min, and a minimum of 50 flies were tested for each genotype and condition.

## Life span assay

Adult virgin flies were collected within 24 hr of eclosion and maintained in groups of 10 in standard culture vials. These vials were kept at 25°C and 60% humidity on a 12:12 hr light-dark cycle. Living flies were transferred to fresh food and their numbers were recorded every 2 days for up to 80 days. A minimum of 50 flies were tested for each genotype.

## Experimental design and statistical analysis

For immunocytochemical experiments, a minimum of 10 larvae were assessed for each genotype and condition. For electrophysiology experiments, data were collected from at least seven independent cells derived from a minimum of four different larvae. Here, each NMJ is consider an n of 1, as each motor axon is restricted to its own muscle cell. For behavioral studies, a minimum of 50 and 100 flies were assessed per genotype, respectively. Detailed information on n used for each experiment is included in the figure legends. Male and female animals were used in comparable numbers except for behavioral and life span assays, in which male adult flies are used. There are no statistical differences in results between the two groups.

All data are shown as mean ± SEM. Data had passed the D'Agostino-Pearson and the Shapiro-Wilk test for normality before being evaluated for statistical significance. Statistical analyses were performed using Prism (7.02, GraphPad), including one-way ANOVA test with Tukey's multiple comparisons, two-way ANOVA test with Tukey's multiple comparisons, and unpaired two-tailed Student's t-test with unequal variance. Results shown represent data pooled from at least two independent experiments performed on larvae or adult flies derived from different crosses. Researchers were blinded to genotype and treatment condition during all experiments and data analyses.

## Acknowledgements

This work was supported by funding from the NIH (NS065053) to AD and the American Heart Association (18PRE34030101) to HL. We thank all members of the DiAntonio lab for helpful discussions and continuous support and Jeanne Nerbonne for her excellent suggestion.

## Additional information

### Funding

| Funder | Grant reference number | Author |
|---|---|---|
| National Institutes of Health | NS065053 | Aaron DiAntonio |
| American Heart Association | 18PRE34030101 | Hailun Li |

The funders had no role in study design, data collection and interpretation, or the decision to submit the work for publication.

### Author contributions

Hailun Li, Conceptualization, Data curation, Formal analysis, Investigation, Methodology, Writing - original draft; Lorenzo Lones, Formal analysis, Investigation; Aaron DiAntonio, Conceptualization, Supervision, Funding acquisition, Project administration, Writing - review and editing

### Author ORCIDs

Hailun Li https://orcid.org/0000-0003-4757-4151
Aaron DiAntonio https://orcid.org/0000-0002-7262-0968

### Decision letter and Author response

Decision letter https://doi.org/10.7554/eLife.62606.sa1
Author response https://doi.org/10.7554/eLife.62606.sa2

## Additional files

### Supplementary files

- Transparent reporting form

### Data availability

All data generated or analyzed during this study are included in the manuscript.

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
