## [Decision Letter]

**Acceptance summary:**

In this paper, the authors examine the function of a salt-inducible-kinase (SIK3) pathway, operating in *Drosophila* glia, in response to K^+^ stress. The authors expand on a pathway they previously described, showing that glial SIK3 functions downstream of neuronal octopamine, a stress ligand. The glial SIK3 pathway is either activated or inhibited, depending on the type of K^+^ stress.

**Decision letter after peer review:**

Thank you for submitting your article "Bidirectional regulation of glial potassium buffering: glioprotection versus neuroprotection" for consideration by *eLife*. Your article has been reviewed by three peer reviewers, and the evaluation has been overseen by Graeme Davis as Reviewing Editor and Richard Aldrich as the Senior Editor. The following individual involved in review of your submission has agreed to reveal their identity: J Troy Littleton (Reviewer #2).

The reviewers have discussed the reviews with one another and the Reviewing Editor has drafted this decision to help you prepare a revised submission.

Summary:

The manuscript by Li and colleagues describes a very nice follow-up analysis to the DiAntonio lab's original discovery of SIK3 AMPK control of a HDAC4-dependent transcriptional program for K^+^ buffering capacity in *Drosophila* wrapping glia. *Drosophila* has been used as a model system to study molecular functions of glia for over two decades. While there are clear differences with vertebrate glia, the similarities appear meaningful and powerful for the discovery of conserved functional principles. In general, glial control of excitability has been a bit of a vague area of research, and precise mechanisms have been difficult to pin down. The work here and in the authors' previous paper does a good job in implicating specific signaling systems in the response. Overall, all of the reviewers are very positive about the work and congratulate the authors for their nice study. There was consensus among the reviewers regarding a few major comments that should be address, listed below. revision.

Essential revisions:

1) It is not clear in the title, or even in the Abstract that the model system being used is *Drosophila*. *Drosophila* is mentioned once in the Abstract with respect to the hyperexcitability model, but not for the rest. Because flies are not necessarily the same as mammals, I think this needs to be made very clear.

2) Given point 1, in the Introduction, the authors really need to explain which fly glia they are looking at, and make the case for similarity with vertebrates based on existing knowledge.

3) In their prior work, the authors demonstrated the SIK3 pathway primarily occurred in wrapping glia that surround axons using wrapping glia-specific GAL4 drivers. That distinction is missing in the current study, as the pan-glial REPO-GAL4 driver was used for all the experiments described here. It seems essential for the authors to define the glial sub-type specific requirements for a number of reasons. For example, the data presented suggest that the blood-brain barrier forming subperineurial glial cells are required given the fast reaction of HDAC4 localization upon bath application of octopamine.

4) Where are the different receptors (Octß1R and OAMB) localized? For both genes Trojan Gal4 insertions (or aGFP converted MiMIC) are available.

---

## [Author Response]

Essential revisions:1) It is not clear in the title, or even in the Abstract that the model system being used is *Drosophila*. *Drosophila* is mentioned once in the Abstract with respect to the hyperexcitability model, but not for the rest. Because flies are not necessarily the same as mammals, I think this needs to be made very clear.

We have now added “*Drosophila*” in prominent locations in the Abstract, Introduction, and Discussion.

2) Given point 1, in the Introduction, the authors really need to explain which fly glia they are looking at, and make the case for similarity with vertebrates based on existing knowledge.

We have added a line to the Introduction highlighting that in the PNS this pathway primarily functions in wrapping glia, made the analogy of wrapping glia to vertebrate non-myelinating Schwann cells, and included a reference for a review of fly glial subtypes, their function, and relationship to vertebrate glia.

3) In their prior work, the authors demonstrated the SIK3 pathway primarily occurred in wrapping glia that surround axons using wrapping glia-specific GAL4 drivers. That distinction is missing in the current study, as the pan-glial REPO-GAL4 driver was used for all the experiments described here. It seems essential for the authors to define the glial sub-type specific requirements for a number of reasons. For example, the data presented suggest that the blood-brain barrier forming subperineurial glial cells are required given the fast reaction of HDAC4 localization upon bath application of octopamine.

We have performed the suggested experiment by expressing FLAG-HDAC4 in wrapping glia. We find that octopamine does promote the nuclear localization of HDAC4 and so inhibit the SIK3 pathway in wrapping glia, and so this new regulatory pathway we describe is functional in wrapping glia. These data are now described in the Results and an accompanying supplementary figure (Figure 3—figure supplement 1).

4) Where are the different receptors (Octß1R and OAMB) localized? For both genes Trojan Gal4 insertions (or aGFP converted MiMIC) are available.

We performed the suggested experiment and found that each receptor was expressed in many but not all peripheral glia. Unfortunately, we could not determine which glia subtypes (such as wrapping glia) express which receptors, because our marker for wrapping glia is also a Gal4 driver (Nrv2-Gal4). Since we could not identify the glia subtypes we did not include these data in the paper.